# The Preparation and Study of Ethylene Glycol-Modified Graphene Oxide Membranes for Water Purification

**DOI:** 10.3390/polym11020188

**Published:** 2019-01-22

**Authors:** Yang Zhang, Lin-jun Huang, Yan-xin Wang, Jian-guo Tang, Yao Wang, Meng-meng Cheng, Ying-chen Du, Kun Yang, Matt J. Kipper, Mohammadhasan Hedayati

**Affiliations:** 1Institute of Hybrid Materials, National Center of International Research for Hybrid Materials Technology, College of Materials Science and Engineering, Qingdao University, Qingdao 266071, China; zyybfq@163.com (Y.Z.); yanxin_2008@126.com (Y.W.); wangyaoqdu@126.com (Y.W.); uniquecheng666@163.com (M.C.); duyingchen@126.com (Y.D.); pepperrose@sohu.com (K.Y.); 2Department of Chemical and Biological Engineering, Colorado State University, Fort Collins, CO 80523, USA; Matthew.Kipper@colostate.edu (M.J.K.); hedayati.che@gmail.com (M.H.)

**Keywords:** graphene oxide, ethylene glycol, membrane, water purification

## Abstract

In this work, graphene oxide (GO)/ethylene glycol (EG) membranes were designed by a vacuum filtration method for molecular separation and water purification. The composite membranes were characterized by scanning electron microscopy (SEM), field emission scanning electron microscopy (FESEM), Fourier transform infrared spectroscopy (FTIR), thermogravimetric analysis (TGA), atomic force microscopy (AFM) and X-ray photoelectron spectroscopy (XPS). The interlayer spacing of GO membranes (0.825 nm) and GO/EG membranes (0.634 nm) are measured by X-ray diffraction (XRD). Using the vacuum filtration method, the membrane thickness can be controlled by selecting the volume of the solution from which the membrane is prepared, to achieve high water permeance and high rejection of Rhodamine B (RhB). The membrane performance was evaluated on a dead-end filtration device. The water permeance and rejection of RhB of the membranes are 103.35 L m^−2^ h^−1^ bar^−1^ and 94.56% (GO), 58.17 L m^−2^ h^−1^ bar^−1^ and 97.13% (GO/EG), respectively. The permeability of GO/EG membrane is about 40 × 10^−6^ L m^-1^ h^−1^ bar^−1^. Compared with the GO membrane, the GO/EG membrane has better separation performance because of its proper interlayer spacing. In this study, the highest rejection of RhB (99.92%) is achieved. The GO/EG membranes have potential applications in the fields of molecular separation and water purification.

## 1. Introduction

Safe and readily available water is an important contributor to public health, whether it is used for drinking, washing, food production, or recreational purposes. Increased demand and declining water resources necessitate advanced water treatment technologies. Recycling of wastewater through advanced wastewater treatment is an area that requires technological advancement [1]. As the problem of fresh water scarcity continues to escalate, water purification becomes one of the most powerful technologies to ensure fresh water supply [2].

Membrane separation technology is considered as one of the main technologies for wastewater treatment and desalination. It is also one of the main ways to obtain a large number of clean fresh water resources. This technology has far-reaching significance for the shortage of fresh water resources. Since the development of high-orientation graphene by physicists Andre Geim and Konstantin Novoselov from University of Manchester in 2004 [3,4], researchers have maintained a high interest in the unique structure and properties of graphene and its derivatives. The detailed study of graphene structure and properties has led to many new proposed applications, including the development of filtration membranes with tunable nanoscale properties [5,6,7,8]. Since graphene materials have been introduced into the field of membrane separation, they have demonstrated outstanding advantages due to its special atomic structure comprised of a single-atom-thick sheet of hexagonally arrayed sp2-bonded carbon atoms [9,10]. In recent years, graphene-based nanofiltration membranes have shown potential applications in filtration, separation and desalination.

Graphene oxide (GO) is an oxidized derivative of graphene. The Lerf-Klinowski model [11,12] describes the presence of oxygen-containing functional groups on GO, including epoxy, hydroxyl, and carboxyl groups. These oxygen functional groups impart important properties. GO has excellent dispersibility in water due to the presence of these hydrophilic groups, which makes GO easy to handle in solution and facilitating film formation. These functional oxygen-containing groups also serve as reactive sites for surface modification reactions. Therefore, GO can be used to develop a series of materials with different surface chemistries, which could significantly improve separation performance.

The ability to manipulate GO in aqueous solutions enables membrane formation by a variety of techniques, including vacuum filtration, layer-by-layer assembly, spray-coating and spin-coating [13,14,15,16]. While these developments have shown potential in the field of membrane separation, an ongoing challenge is to optimize the trade-off between water permeance and solute rejection. J. Abraham et al. controlled the layer spacing of GO membrane and achieved accurate and tunable ion screening through physical limitations. They adopted a simple, scalable method to obtain a graphene-based membrane with limited swelling, which showed 97% NaCl rejection [17]. Liang Chen et al. changed the interlayer spacing of GO films by adding cations to GO to improve membrane performance [18]. Huiyuan Liu et al. manufactured a free-standing ultrathin reduced GO (rGO) membrane by a simple and effective method. The ultra-thin and strong rGO membrane not only overcomes the key challenge of internal concentration polarization, but also has high rejection of organic and inorganic species [19]. As this body of work demonstrates, tuning the chemistry and structure of GO membranes can result in improved membrane performance for a variety of water treatment applications. In particular, modification with small organic molecules has attracted interest for a wide range of applications [20], because they can change the layer spacing of GO membranes.

Ethylene glycol (EG) is a small organic molecule with good water solubility. The bishydroxy structure of EG can be covalently attached to the surface of GO, without affecting the water solubility of GO. We therefore propose modifying GO with EG to changing the layer spacing of GO membranes.

In our previous work, we have reported graphene-based hybrid materials such as GO/rare-earth materials [21], graphene/silver hybrid membranes [22], GO/polyacrylamide composite membranes [23] and EO-gold nanoparticles membranes [24], and discussed their application in water purification. In this paper, EG was grafted onto the surface of GO, altering the distance between layers of GO to achieve improvements in both water permeance and solute rejection of GO membranes. The GO/EG composite membranes were prepared by vacuum filtration, which is conducted in aqueous solution without any organic solvents. The layer-by-layer stacking structure can be observed clearly by scanning electron microscopy (SEM) and field emission scanning electron microscopy (FESEM), and the interlayer spacing of the membranes was measured by X-ray diffraction (XRD). For comparison, we studied water permeance and rejection performance of both GO and GO/EG membranes.

## 2. Materials and Methods

### 2.1. Materials

Natural graphite (Qingdao Tianheda Graphite Co., Ltd., Qingdao, China), potassium persulfate (K_2_S_4_O_8_), sulfuric acid (H_2_SO_4_, 98.0%), potassium permanganate (KMnO_4_, 99%) and phosphorus pentoxide (P_2_O_5_) (Zhiyuan Chemical reagent Co., Ltd., Tianjin, China) were used to prepare GO. Hydrogen peroxide (H_2_O_2_, 30%) and hydrochloric acid (HCl, 37%) were used to wash the reaction solution. Barium chloride (BaCl_2_) was used to check acid ions. Ethylene glycol (C_2_H_6_O_2_) was used to modify GO. These chemicals were analytical grade and provided by Beijing Chemical Co., Ltd., Beijing, China.

### 2.2. Preparation of GO nanosheets and GO/EG

There are many methods for the preparation of GO [19,25,26,27]. In this study, GO nanosheets were synthesized from natural graphite according to a modified Hummers’ method [28]. The GO/EG composite is formed by attaching EG to the GO surface via an esterification reaction. First, 50 mg of GO was dispersed in 100 mL distilled water followed by sonication for 30 min. Then, the GO solution was transferred to a flask. Lastly, EG (10 mL) was added to the GO solution and stirred at 40 °C for 8 h to form a homogeneous GO/EG composite solution. The reaction process of GO nanosheets and GO/EG composite as shown in Figure 1 (①, ②).

### 2.3. Fabrication of GO and GO/EG composite membranes

The GO and GO/EG composite membranes were prepared by vacuum filtration under a transmembrane pressure of 1 bar (0.1 MPa) and the temperature of solutions was controlled at 25 °C, as shown in Figure 1 (③, ④) and Appendix A [13]. The solution of GO and GO/EG was filtered through a 0.22 μm mixed cellulose membrane (Φ 50mm, Shanghai Xingya purifying material factory, Shanghai, China) to produce a membrane. The thickness of the GO and GO/EG membrane can be easily controlled by changing the volume of the GO or GO/EG solution that is filtered through the membrane. The obtained membranes were dried in vacuum at 40 °C before use. The effective diameter of each membrane was 4 cm as can be seen in Appendix A.

### 2.4. Characterization of GO and GO/EG composite membranes

To understand the surface and cross section structure of the GO and GO/EG composite membranes, the prepared membranes were characterized by scanning electron microscopy (SEM; JEOL 6460, Hitachi Limited, Tokyo, Japan) and field emission scanning electron microscopy (FESEM, JSM-7500F, Hitachi Limited, Tokyo, Japan). Fourier transform infrared spectroscopy (FTIR; MAGNA-IR 550, Varian, Inc., Palo Alto, CA, USA) and X-ray photoelectron spectroscopy (XPS; AXIS ULTRA DLD, KRATOS, London, UK) spectra of the GO and GO/EG composite membranes were recorded to analyze the functional groups. Thermal gravimetric analysis (TGA; TA-Q50, Waltham, MA, USA) was carried out in N_2_ atmosphere to analyze the thermal properties of the GO and GO/EG composite. X-ray diffraction (XRD, DMAXRB-III, Rigaku, Kyoto, Japan) was used to measure the layer spacing of the GO and GO/EG composite membranes. The effect of GO modification on surface roughness was studied by analyzing the membranes surface using Atomic Force Microscopy (AFM; Bruker Bioscope Resolve, Akishima, Japan).

### 2.5. GO and GO/EG composite membranes permeance and rejection test

The performance of the membrane was examined by measuring the pure water permeance, and by measuring the rejection of RhB, using a solution of 0.02 g/L RhB. As shown in the Appendix A. In this work, the common dye RhB is used as a model for organic pollutants to evaluate membrane performance. The water permeance values were calculated using the equation:(1)$$K=\frac{V}{S\times t\times P}$$
***V*** is the volume of water, ***S*** is the effective filtration area of the membranes (1.256 × 10^−3^ m^2^), ***t*** is the filtration time, ***P*** is the transmembrane pressure (0.1 MPa).

The permeability was defined in order to better describe the relationship between water permeance and membrane thickness. The permeability ***I***, using the following equations:(2)I=K×d
***K*** is the water permeance, ***d*** is the membrane thickness.

The rejection was characterized by using RhB solution. Then, the concentration of the RhB solution is proportional to the absorbance, ***A***, according to Beer–Lambert law:(3)A=k×l×c
where ***k*** is the molar absorptivity coefficient, ***l*** is the path length, and ***c*** is the solution concentration.

The rejection was calculated as the % change in solution concentration. The rejection ***R***, using the following equations:(4)$$R=\frac{A_{1}-A_{2}}{A_{1}}\times 100\%$$
where ***A***_1_ is the absorbance of original RhB solution, ***A***_2_ is the absorbance of the RhB passing through the membrane.

#### Calculation of Membrane Interlayer Spacing

The diffraction angles of GBMs were obtained by XRD characterization, and the interlayer spacing was calculated by the Bragg’s law:(5)2dsinθ=λ

***d*** is the lattice spacing, ***θ*** is the diffraction angle, ***λ*** is the wavelength.

## 3. Results

### 3.1. SEM analysis

To investigate the structure of GO and GO/EG membranes, the cross-section and morphology of the membranes were observed. Figure 2a,b show SEM images of the surface of the GO and GO/EG membranes. From Figure 2a,b, we can clearly observe that the surface of the GO/EG membrane has more wrinkles than the surface of the GO membrane. This shows that the GO/EG membrane has a larger surface roughness than the GO membrane, which can also be proved in the AFM test (Figure 9), below. Figure 2c,d show FESEM images of the surface and the cross-section of the GO/EG composite membrane, respectively. The FESEM images display a wrinkled surface and stacked layered structure. The nano-sized channels between the layers can allow small molecules (such as water molecules) to pass through, enabling separation at a molecular scale.

### 3.2. Thermal gravimetric analysis

Figure 3 shows the TGA (a) and DTG (b) curves of GO and GO/EG. By comparing the two curves of TGA and DTG, we can obtain that the GO and GO/EG have three mass loss processes [29], but the mass loss rate of the GO and GO/EG are different. The GO membrane has a process of weight loss below 105 °C, which corresponds to 14.6% of the total mass and corresponds to the evaporation of moisture contained in GO sample. The 26.6% weight loss occurs between 105 °C and 240 °C due to the decomposition of oxygen-containing groups on the surface of GO. The last stage occurs between 240 °C and 800 °C. This mass loss is due to sublimation of the carbon skeleton, resulting in a loss of 8% of the total initial mass. The first stage of GO/EG weight loss also occurs below 105 °C, but the weight loss is only 4.8% of the total mass. The second stage occurs between 105 °C and 240 °C. In this stage, the weight loss is 15.7% of the total mass, which is due to decomposition of functional groups on the GO surface. The 18.4% weight loss occurs between 240 °C and 800 °C, due to the sublimation of the carbon skeleton.

By comparing the TGA and DTG curves of GO and GO/EG, we find that the GO/EG has better thermal stability than the GO, evidenced by the higher thermal decomposition temperature of ester groups than carboxyl, hydroxyl and epoxy groups. Therefore, we can conclude that EG is successfully attached onto the surface of GO.

### 3.3. FTIR analysis

Figure 4a displays the FTIR spectra of GO and GO/EG, which provides information about detailed chemical bonds and functional groups. By comparing the infrared spectra of GO and GO/EG, we observe that curve B has two more peaks at 2940 cm^−1^ and 2870 cm^−1^ [30] than curve A, which are due to the stretching vibration of C–H in EG. The peaks of curve B at 1730 cm^−1^ and 1624 cm^−1^ [31] are significantly reduced, because the carboxyl group on the GO surface is substituted [32]. These spectra confirm that EG is attached to the surface of GO.

### 3.4. XPS analysis

Figure 4b shows the survey scans of GO membrane and GO/EG membrane. Figure 4c,d shows high-resolution XPS peaks of the GO membrane (O 1s), and the GO/EG membrane (O 1s). The high-resolution peak shown in Figure 4c, the O 1s peak region of the GO membrane has two component peaks: C=O or O–C=O (532.2 eV) and C−O−C or C−O−H (533.8 eV) [33]. This is because the surface of the GO has a large number of oxygen-containing groups such as a hydroxyl group and a carboxyl group. Compared to Figure 4c,d there is an additional peak corresponding to C−O−C=O (533.7 eV) [34], which is due to the grafting of EG onto the surface of the GO to form new ester groups.

### 3.5. Water permeance tests and permeability

To investigate the permeance of GO and GO/EG membranes and the effect of membrane thickness on pure water permeance, we measured the pure water permeance of several different thickness membranes. The results are shown in Figure 5. In the vacuum filtration method used to form the membranes, we can control the thickness of the membrane by controlling the amount of aqueous solution used to prepare the membrane, because both GO and GO/EG can form stable aqueous solutions. The water permeance is negatively correlated with the thickness of the membrane. When the volume of solution is the same, the pure water permeance of the GO membrane is higher than the pure water permeance of the GO/EG membrane, which is determined by the layer spacing of the membrane. We obtained the GO/EG membrane with smaller layer spacing, because a large number of oxygen-containing groups on the GO surface are substituted at the same time, as shown in Figure 8, below. EG modified on the surface of GO partially enters the channels formed by GO sheet stacking, thereby reducing the water permeance of GO/EG membranes. The results of pure water permeance of GO membrane and GO/EG membrane also agree with the results of XRD shown in Figure 7e, below.

The permeability was defined in order to better describe the relationship between water permeance and membrane thickness. The permeability is obtained by multiplying the permeance by the membrane thickness. When the volume of solution is the same, the permeability of the GO membrane is higher than that of the GO/EG membrane as shown in Table 1. Because the water permeance is negatively correlated with the rejection, so, the GO/EG membrane has higher rejection than the GO membrane. The results also agree with the rejection test shown in Figure 6, below. As shown in the Table 1, the permeability of the GO/EG membranes is approximately constant with respect to membrane thickness, as expected. However, for the GO membranes, the permeability decreases as the membrane thickness increases, suggesting some non-uniform structure or other non-ideal behavior in the GO membrane. In addition, we can ignore the influence of the membrane resistance of the support membrane because the pores of the support membrane are large, as shown in Appendix A.

### 3.6. Rejection tests

We used an RhB aqueous solution (0.02 g/L) to test the rejection of the GO and GO/EG membranes for RhB. The RhB concentration in the filtrate and unfiltered solution was measured by UV-Vis spectroscopy. As shown in Figure 6, we can find a positive correlation between the rejection and the thickness of the membrane. For GO and EG/GO membranes prepared from the same volume of solution, the rejection of GO/EG membrane is significantly higher than the rejection of the GO membrane for RhB, due to the different layer spacing of the membrane. The high rejection GO/EG membrane may also make these membranes suitable for other applications, such as seawater desalination and wastewater treatment.

## 4. Discussion

The layered structure of the membranes is seen in Figure 7a,c and (Appendix A). The separation principle of the membranes is illustrated schematically in Figure 7b,d: water molecules can pass through the membranes via the spacing between the layers, while macromolecules or other impurities cannot pass between the membrane layers [2,35]. We used XRD to characterize the layer spacing (dry state) of GO and GO/EG membranes. As shown in Figure 7e, the diffraction peak of the GO membrane is observed at 2*θ* = 10.72°, corresponding to the crystal plane (001) of GO [36]. The layer spacing of GO is 0.825 nm (d1) from Figure 7b, which is calculated by the Bragg equation [22,37,38]. The characteristic diffraction peak moves to 2*θ* = 13.96°, following the grafting of EG onto the surface of GO. As seen in Figure 7d, the corresponding calculated interlayer spacing of the GO/EG membrane is 0.634 nm (d2). The GO/EG membrane has smaller interlayer spacing than GO membrane. As shown in Figure 8, this is because the EG is attached to the surface of the GO, so that the thickness of the single-layer GO nanosheet is increased, which causes the layer spacing of the GO/EG to become smaller. The XRD results are consistent with the results of the rejection tests (Figure 6).

The surface roughness of the membranes were measured by AFM shown in Figure 9 and Table 2. The roughness average (*Ra*) and root mean square roughness (*Rq*) for GO membrane (Figure 9a,b) and GO/EG membrane (Figure 9c,d) were obtained in a scan of 10 μm × 10 μm. The *Ra* and *Rq* of GO/EG membrane are significantly higher than the *Ra* and *Rq* of the GO membrane, which is due to the grafting of ethylene glycol onto the surface of the GO membrane. This is consistent with the SEM analysis (Figure 2).

## 5. Conclusions

In this study, we developed novel graphene-based membranes (GO/EG membranes) for molecular separation and water purification. The selective permeation membrane was prepared by a vacuum filtration method. These membranes are permeable to water, but other solutes can be rejected by size exclusion. We have confirmed that the layer spacing of GO/EG membranes are smaller than the original membranes. Therefore, the GO/EG membranes have a higher rejection than the GO membranes. The water permeance is negatively correlated with the thickness of the membrane and the rejection of a model organic dye molecule is positively correlated with the thickness of the membrane. The GO/EG membranes can be used for water purification due to good water permeance and high rejection of large sized molecules. In addition, GO/EG membranes have potential applications for molecular separation and water purification.

## Figures and Tables

**Figure 1 polymers-11-00188-f001:**
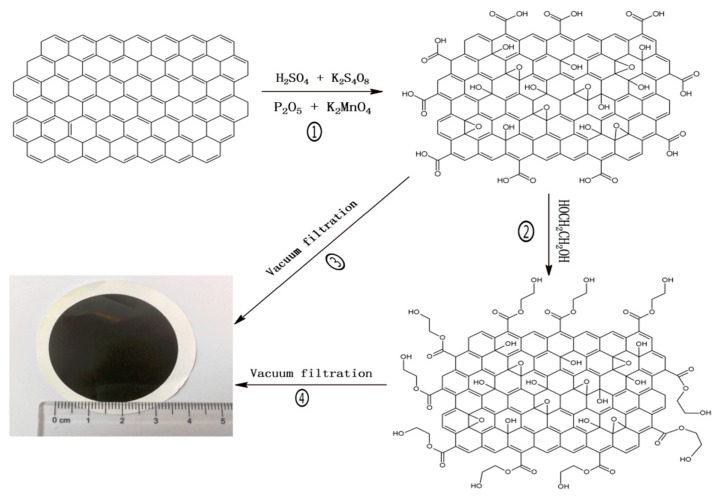
(①, ②) shows the reaction process of GO nanosheets and GO/EG composite; (③, ④) shows the membrane preparation process of GO and GO/EG.

**Figure 2 polymers-11-00188-f002:**
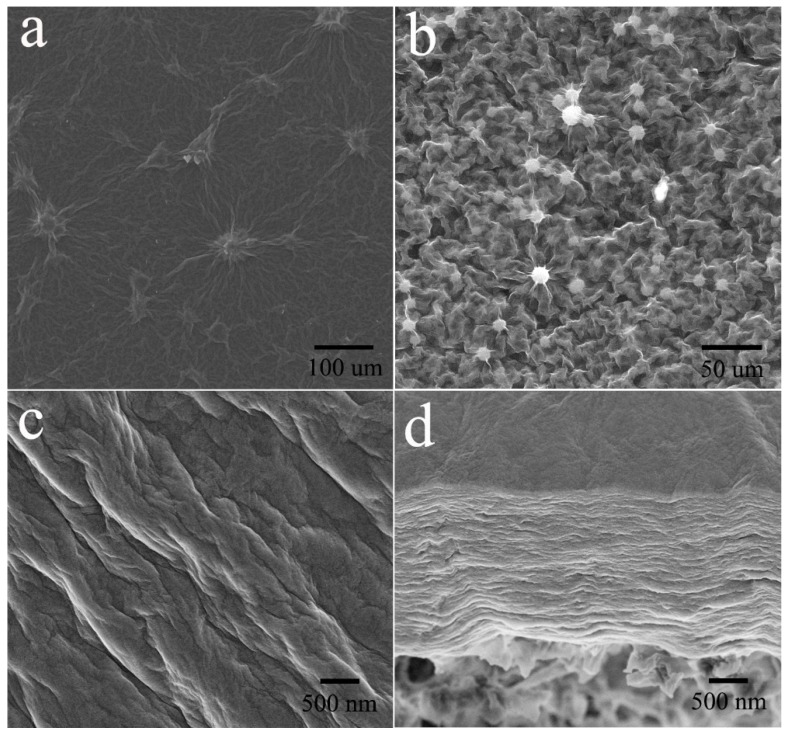
SEM (**a**) images of the surface of GO membrane. (**b**) SEM images of the surface of GO/EG membrane. FESEM images of (**c**) the surface and (**d**) the cross section of GO/EG composite membrane.

**Figure 3 polymers-11-00188-f003:**
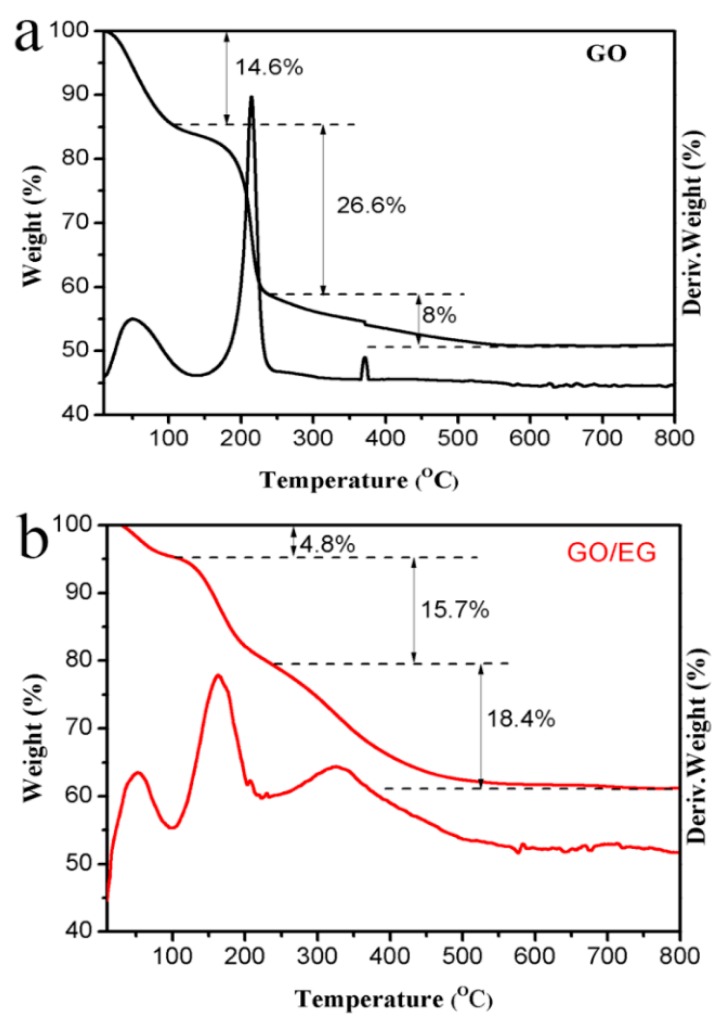
(**a**) TGA curve and (**b**) DTG curve of GO and GO/EG under N_2_ atmosphere (10 mL/min) at a heating rate of 10 °C/min.

**Figure 4 polymers-11-00188-f004:**
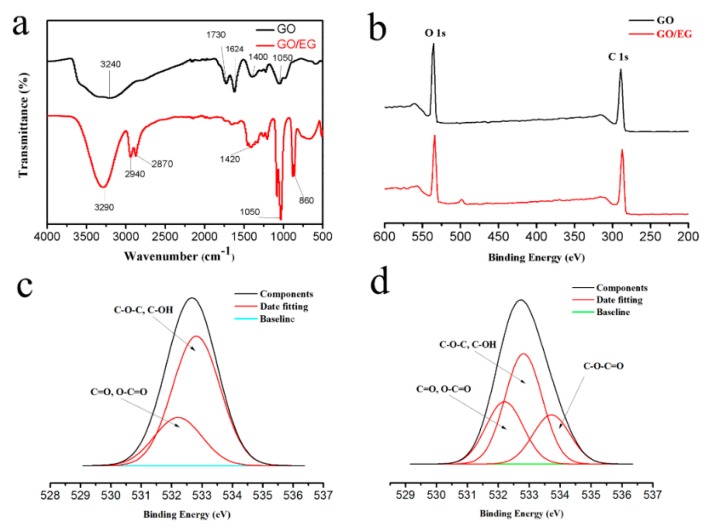
(**a**) FTIR spectra of GO and GO/EG composite; (**b**) survey scans of GO and GO/EG membranes in the spectral region 200–600 eV; (**c**) O 1s region of GO membrane; (**d**) O 1s region of GO/EG membrane.

**Figure 5 polymers-11-00188-f005:**
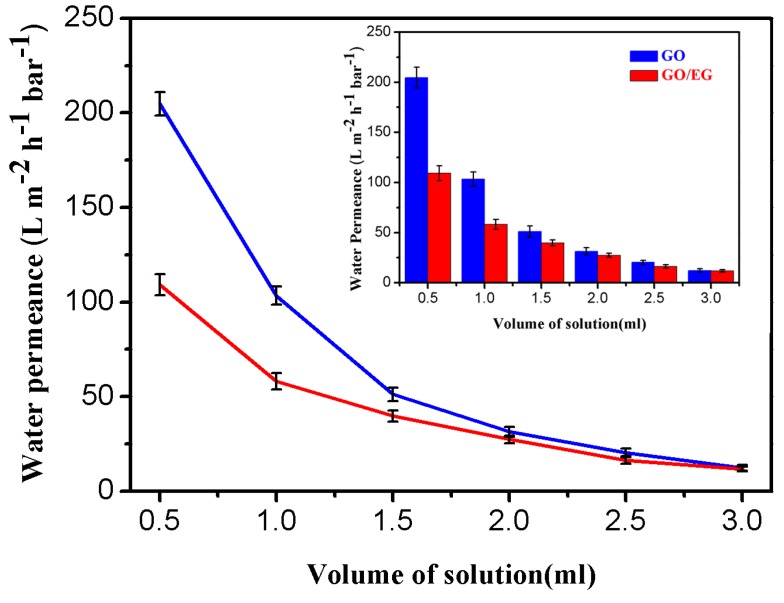
The water permeance of GO and GO/EG membranes, as a function of the amount of GO and EG/GO used to prepare the membrane.

**Figure 6 polymers-11-00188-f006:**
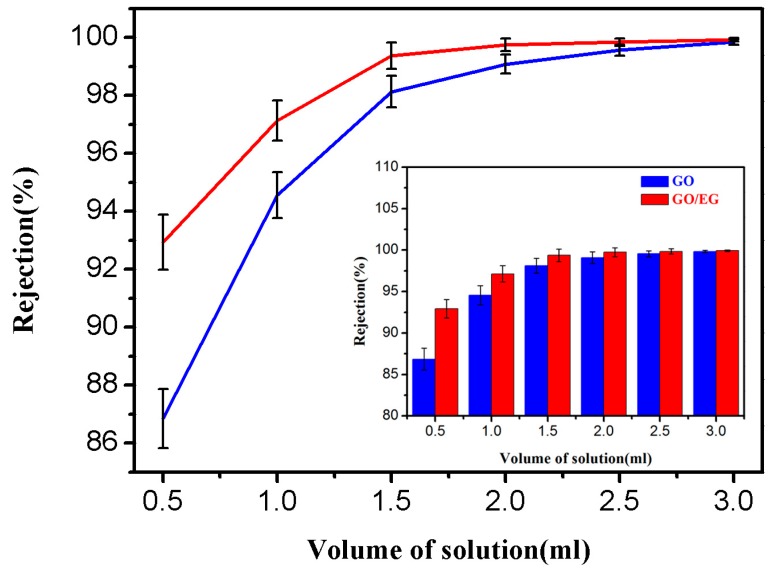
The rejection rate of RhB of GO and GO/EG membranes.

**Figure 7 polymers-11-00188-f007:**
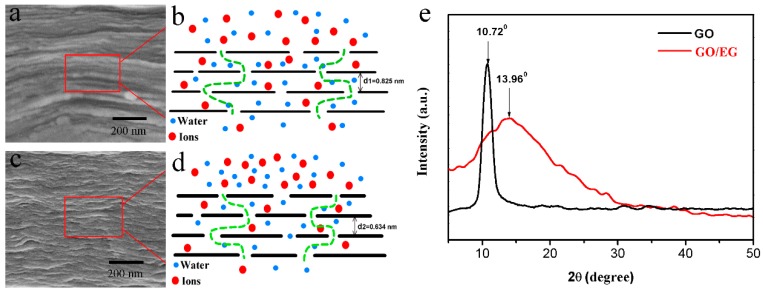
(**a,c**) FESEM images of the cross section of GO and GO/EG composite membrane, (**b,d**) Schematic diagram for the permeation through GO and GO/EG membrane laminates, (**e**) The XRD pattern of GO and GO-EG composite.

**Figure 8 polymers-11-00188-f008:**
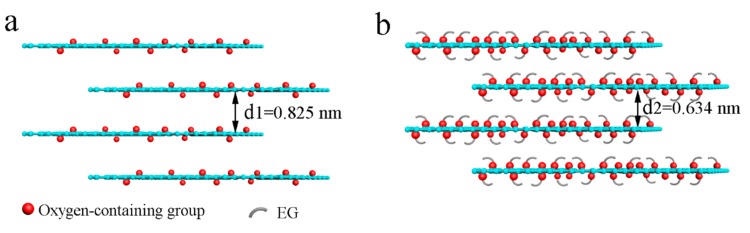
Schematic diagram of EG modified GO.

**Figure 9 polymers-11-00188-f009:**
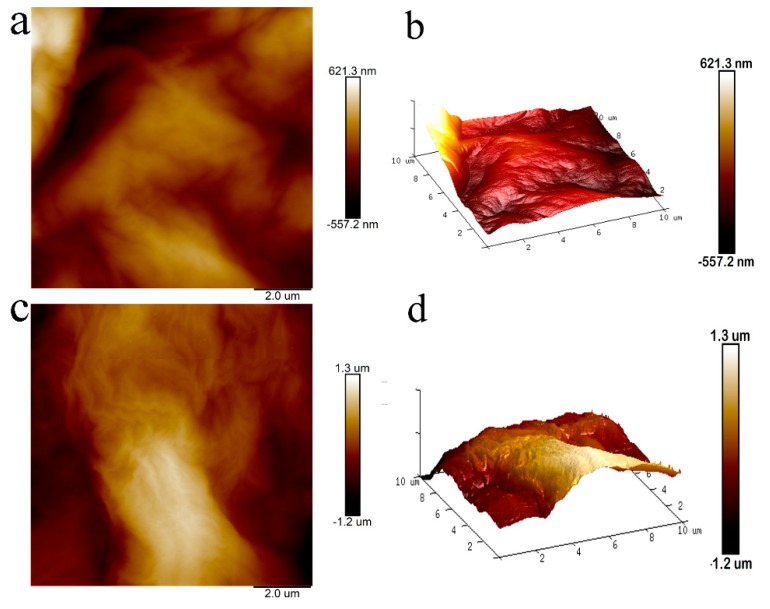
Two-dimensional and three-dimensional AFM images of GO membrane (**a,b**) and GO/EG membrane (**c,d**).

**Table 1 polymers-11-00188-t001:** The permeability of GO and GO/EG membranes.

VOS	GO MT	GO P	GO IP	GO/EG MT	GO/EG P	GO/EG IP
ml	10^−6^ m	L m^−2^ h^−1^ bar^-1^	10^−6^ L m^−1^ h^−1^ bar^−1^	10^−6^ m	L m^−2^ h^−1^ bar^−1^	10^−6^ L m^−1^ h^−1^ bar^−1^
0.5	0.36 ± 0.03	205 ± 10	74 ± 4	0.35 ± 0.03	110 ± 5	39 ± 2
1.0	0.72 ± 0.03	103 ± 5	74 ± 4	0.70 ± 0.03	58 ± 3	41 ± 2
1.5	1.08 ± 0.03	51 ± 3	55 ± 3	1.05 ±0.03	40 ± 2	42 ± 2
2.0	1.44 ± 0.03	31 ± 2	45 ± 2	1.40 ± 0.03	27 ± 1	38 ± 2
2.5	1.80 ± 0.03	24 ± 1	43 ± 2	1.75 ± 0.03	21 ± 1	37 ± 2
3.0	2.16 ± 0.03	19 ± 1	41 ± 2	2.10 ± 0.03	18 ± 1	38 ± 2

**VOS**: volume of solution; **MT**: membrane thickness; **P**: permeance; **IP**: permeability.

**Table 2 polymers-11-00188-t002:** Roughness parameter of the surface of GO and GO/EG membrane.

Sample	Roughness Average (Ra)	Root mean Square Roughness (Rq)
GO	136 ± 5nm	169 ± 5nm
GO/EG	336 ± 5nm	407 ± 5nm

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
