# Peer review of "The Preparation and Study of Ethylene Glycol-Modified Graphene Oxide Membranes for Water Purification"

_polymers, 2019, doi:10.3390/polym11020188_

Round 1

Reviewer 1 Report

The authors present a study on the preparation of GO and GO modified with ethylene glycol membranes for filtration application. The GO and modified GO membranes were characterized by SEM microscopy, TGA analysis, FTIR and XPS spectroscopy. The permeability and rejection properties were tested with water and RhB molecules.

While the study is interesting, some aspects are not clear and the authors are invited to address them in order to improve their manuscripts. The following comments are appended to aid the authors in revising the manuscript:

1. The characterization results present the GO membrane confronted with a modified GO membrane. However no details regarding the characteristics of the modified GO membrane are given so as to help readiness and clarify the properties vs the preparation aspects. Please add such details in all the characterization results (SEM, FTiR, etc.)

2. In the experimental section, there is ambiguity in preparation method. It appears the membranes are supported by a ‘mixed cellulose membrane’. How were the GO membranes achieved exactly? Are they free standing? How were they detached from the supporting membrane? Do they maintain the supporting membrane? – then how was this taken into account for the permeability and rejection tests or the characterization such as TGA etc?

3. For improved clarity it is advised to add the absorbance spectra correspsonding to RhB rejection tests with the modified GO membrane.

Author Response

First of all, we are very grateful to reviewers for your valuable comments and suggestions. The following, we will respond one by one:

Reviewer 1

The authors present a study on the preparation of GO and GO modified with ethylene glycol membranes for filtration application. The GO and modified GO membranes were characterized by SEM microscopy, TGA analysis, FTIR and XPS spectroscopy. The permeability and rejection properties were tested with water and RhB molecules.

While the study is interesting, some aspects are not clear and the authors are invited to address them in order to improve their manuscripts. The following comments are appended to aid the authors in revising the manuscript:

Comment 1. The characterization results present the GO membrane confronted with a modified GO membrane. However no details regarding the characteristics of the modified GO membrane are given so as to help readiness and clarify the properties vs the preparation aspects. Please add such details in all the characterization results (SEM, FTiR, etc.)

ResponsesThis is a good suggestion. We have added details in the characterization results. As can be seen in the manuscript in page 4 line 152 to page 4 line 155 (SEM). We have added details of “From Fig. 2a and Fig.2b, we can clearly observe that the surface of the GO/EG membrane has more wrinkles than the surface of the GO membrane. This shows that the GO/EG membrane has a larger surface roughness than the GO membrane, which can also be proved in the AFM test (Fig.9), below.” The GO/EG composite material rather than GO/EG composite membrane is characterized by TGA and FTIR. These characterizations mainly prove that EG is attached to GO. The details are as followsFrom the described in the manuscript in page 5 line 165 to page 5 line174, we find that the GO/EG has better thermal stability than the GO. Therefore, we can conclude that EG is successfully attached onto the surface of GO; From the described in the manuscript in page 6 line 184 to page 6 line187, we also can conclude that EG is successfully attached onto the surface of GO by comparing the infrared spectra of GO and GO/EG.

Comment 2. In the experimental section, there is ambiguity in preparation method. It appears the membranes are supported by a ‘mixed cellulose membrane’. How were the GO membranes achieved exactly? Are they free standing? How were they detached from the supporting membrane? Do they maintain the supporting membrane? – then how was this taken into account for the permeability and rejection tests or the characterization such as TGA etc?

Responses: These are very good questions. In this article, we used a dead-end filtration device, as shown in Fig. S1 in the manuscript in page 3 line 102. The GO/EG composite membranes were prepared by vacuum filtration and we can control the thickness of the membrane by controlling the amount of aqueous solution. First, the GO/EG membranes can free standing. However, the GO/EG membranes are ultra-thin and fragile, which isn’t conducive to the performance test of the GO/EG membranes. Therefore, the GO/EG membranes and the support membranes are integral in the performance test. From Fig. S4 and Table S1 in the manuscript in page 8 line 228 to page 8 line 229, we can obtain that the permeability of the support membrane is much larger than the permeability of the GO/EG membrane, which has a negligible effect on the performance of the GO/EG membrane. And we wish that all these answers will meet your requirements.

Fig. S1 schematic diagram of vacuum membrane forming device.

Comment 3. For improved clarity it is advised to add the absorbance spectra correspsonding to RhB rejection tests with the modified GO membrane.

Responses: Thank you very much for your valuable suggestion. We have added the absorbance spectra corresponding to RhB to clarity the rejection tests of the GO/EG membranes in the manuscript in page 4 line 124. As shown in the Fig. S3, we can observe that the absorption wavelength of RhB is 554 nm. As shown in the Fig. S3 (a), we can observe that the absorbance of the original RhB solution is much higher than the absorbance of the filtrate (0.5, 1, 1.5, 2, 2.5 and 3 represent the RhB filtrate of the corresponding GO/EG membrane, respectively.). We can obtain that GO/EG membranes have a significant rejection effect of RhB. As shown in the Fig. S3 (b), we can find a positive correlation between the rejection and the thickness of the GO/EG membrane.

Fig. S3 the absorbance spectra corresponding to RhB of GO/EG membrane.

Reviewer 2 Report

- In general, the English language, sentence structure and grammar have to be improved, in particular in section 2. 

- Please add the describe the significance and the main merits / expected outcomes of introducing EG into GO membranes in the introduction (section 1) and discussion (section 2).

- Lines 92-94: This conclusion has to be supported by results and the quality of the pics are not the best, I can not see such stacked layered structure. Also, TEM should be also performed!

- Fig.1 (a) and (b), Please explain clearly the difference in the surface morphology due to the incorporation of EG.

- Lines 123-125: The peak at 1420 cm-1 is referring to C-O (in phenol) not hydroxyl groups!! Usually, acid O-H is between 2500-3400 and alcoholic O-H is in range of 3200-3400. Please recheck this part.

- Section 2.5 (Lines 157-159) and Section 4.5 (Lines 272-273 & Equation (2)): The authors are mixing the scientific terms. The terms permeability and intrinsic permeability should be corrected to permeance and permeability. Permeance is the relevant term correlating the barrier thickness and permeability. Please recheck and correct.

- From the membrane point of view: Pore size distribution, porosity and / or MWCO. I do not find this in the current work.

- Section 2.6: the rejection results can not be understood without evaluating the MWCO of the EG/GO membranes.

-Lines: 205-212: In your point of view: What is the influence of surface roughness on the mass transport inside EG/GO and GO membranes?

- Table 3 is misleading and should be deleted, rejection of dyes should not be compared with rejection of other solutes, e.g., salts and proteins!

Author Response

- In general, the English language, sentence structure and grammar have to be improved, in particular in section 2. 

Responses: In the first place, we expressed our great gratitude for the reviewer’s responsibility. In order to improve the English writing of this paper, we have asked for an expert who is a native English-speaker. Now we have recheck the entire manuscript and corrected some errors with our best efforts. As following:

1)      We replaced “solubility” with “dispersibility” in the manuscript in page 2 line 50.

2)      We replaced “The degree of surface undulation of the surface shown in Fig. 1(b) is smaller than that shown in Fig. 1(a), which is due to the reduction of the oxygen-containing groups on the GO surface.” with “From Fig. 2a and Fig.2b, we can clearly observe that the surface of the GO/EG membrane has more wrinkles than the surface of the GO membrane. This shows that the GO/EG membrane has a larger surface roughness than the GO membrane, which can also be proved in the AFM test (Fig.9), below.” in the manuscript in page 4 line 152 to page 4 line 155.

3)      We replaced “By comparing the two curves of TGA and DTG, we can obtain the three mass loss processes of GO and GO/EG…” with “By comparing the two curves of TGA and DTG, we can obtain that the GO and GO/EG have three mass loss processes…in the manuscript in page 5 line 164.

4)      We replaced “The GO membrane has a process of weight loss below 105 °C. This mass loss corresponds to 14.6% of the total mass…” with “The GO membrane has a process of weight loss below 105 °C, which corresponds to 14.6% of the total mass…” in the manuscript in page 5 line 166.

5)      We have deleted “In addition, the peak at 1420 cm-1 of curve B is significantly stronger than that of curve A, which is due to the introduction of hydroxyl groups.” in the manuscript in page 6 line 187.

6)      We replaced In the Table 1 that the permeability of the GO/EG membranes is…” with “As shown in the Table 1, the permeability of the GO/EG membranes is…” in the manuscript in page 8 line 223 to page 8 line 225.

7)      We replaced “atomic force microscopy” with “AFM” in the manuscript in page 10 line 263.

- Please add the describe the significance and the main merits / expected outcomes of introducing EG into GO membranes in the introduction (section 1) and discussion (section 2).

Responses: Thank you for your valuable suggestion. The main merits of introducing EG into GO have been mentioned in the manuscript in page 2 line 70 to page 2 line 72. As follows “Ethylene glycol (EG) is a small organic molecule with good water solubility. The bishydroxy structure of EG can be covalently attached to the surface of GO, without affecting the water solubility of GO.” The expected outcomes of introducing EG into GO is to change the layer spacing of GO membranes to gain smaller layer spacing of GO/EG membrane, which also agree with the results of rejection test shown in Fig. 6 and XRD shown in Fig. 7 in the manuscript in page 9 line 241 and in page 9 line 257, respectively.

- Lines 92-94 (156-158 in the new version): This conclusion has to be supported by results and the quality of the pics are not the best, I can not see such stacked layered structure. Also, TEM should be also performed!

Responses: This is a good valuable suggestion. First of all, the GO/EG composite membranes were prepared by vacuum filtration. Therefore, the cross section of the membranes should be a stacked layer-by-layer stacked structure, which can be confirmed form the FESEM image in the manuscript in page 5 line 159. As we described in the manuscript in page 9 line 257 (Fig. 7 b,d), water molecules can pass through the membrane by the channel between the layers. Then, the reviewer can not see such stacked layered structure because the combined Fig. 2d is too small. From the original Fig. R2, below, we can clearly observe the layered cross section of the membrane. Finally, we have also done TEM characterization, as shown in the Fig. R3, below. Although we can observe the wrinkles on the GO and GO/EG sheets from the Fig. R3, we can’t clearly observe the obvious changes of introducing EG into GO surface. Therefore, we have not added this part to the manuscript.

Figure. R2 the cross section of GO/EG composite membrane.

Figure. R3 TEM images of a (GO) and b (GO/EG).

- Fig.1 (a) and (b), Please explain clearly the difference in the surface morphology due to the incorporation of EG.

Responses: Thank you for your valuable suggestion. We have added details of “From Fig. 2a and Fig.2b, we can clearly observe that the surface of the GO/EG membrane has more wrinkles than the surface of the GO membrane. This shows that the GO/EG membrane has a larger surface roughness than the GO membrane, which can also be proved in the AFM test (Fig.9), below.” to clearly the difference in the surface morphology due to the incorporation of EG in the manuscript in page 4 line 152 to page 4 line 155..

- Lines 123-125 (187 in the new version): The peak at 1420 cm-1 is referring to C-O (in phenol) not hydroxyl groups!! Usually, acid O-H is between 2500-3400 and alcoholic O-H is in range of 3200-3400. Please recheck this part.

Responses: Yes, you are absolutely right, this is our mistake. We have recheched this part and also retrieved some relevant literatures. The peak at 1420 cm-1 is referring to C-O (in phenol) not hydroxyl groups. We have deleted “In addition, the peak at 1420 cm-1 of curve B is significantly stronger than that of curve A, which is due to the introduction of hydroxyl groups.” in the manuscript. Thank you very much again.

- Section 2.5 (Lines 157-159) and Section 4.5 (Lines 272-273 & Equation (2)): The authors are mixing the scientific terms. The terms permeability and intrinsic permeability should be corrected to permeance and permeability. Permeance is the relevant term correlating the barrier thickness and permeability. Please recheck and correct.

Responses: Thank you for your valuable suggestion. We have recheck the entire manuscript and corrected.

- From the membrane point of view: Pore size distribution, porosity and / or MWCO. I do not find this in the current work.

Responses: This is a valuable question. In order to better explain this question, we have retrieved a large number of relevant literatures. It is found that porosity is one of the important parameters of the filteration membrane. The porosity is usually calculated in according to dry-wet weight method and density method. Unfortunately, there is no uniform method to determine the porosity of graphene-based nanofiltration membranes have been reported in the literature. As far as we know, the theoretical porosity of graphene oxide membrane is only 50% [14]. We have try our best to calculate the porosity of GO/EG membrane according to dry-wet weight method through the formula:

Where ρ is the density of ethanol; A is the effective area of the membrane; L is the thickness of the membrane; W1 is the wet membrane quality; W2 is the dry membrane quality.

As a result, the porosity of GO/EG membrane is about 54.83%, which indicated the porosity of GO/EG membrane is higher than that of GO membrane.

Since there is no uniform method to determination the porosity of graphene-based membrane, we have not added this part to the manuscript. We believe that the determination of the porosity of graphene-based nanofiltration membrane is a topic of great research value, and we will conduct systematic research in this direction.

[14] Nair, R.R.; Geim, A.K. Unimpeded Permeation of Water Through Helium-Leak-Tight Graphene-Based Membranes. Science 2012, 335, 442-444.

- Section 2.6: the rejection results can not be understood without evaluating the MWCO of the EG/GO membranes.

Responses: This is a valuable question. MWCO is also one of the important parameters of the filteration membrane. In this paper, there have twelve different thickness membranes (six GO membranes and six GO/EG membranes). Therefore, there are twelve MWCO corresponding. Of course, the MWCO of these membranes will not exceed 500 according to the MWCO of RhB.

However, we did not add this part of the experiment and analysis to the manuscript. If we add the analysis, there will be more variable effects. But we are very grateful for your valuable comments, and your comments have pointed out the direction for our next plan: Study on the retention for different dyes by the same membrane.

-Lines: 205-212 (263-268 in the new version): In your point of view: What is the influence of surface roughness on the mass transport inside EG/GO and GO membranes?

Responses: This is a very good question. I am very sorry that the surface roughness has no direct influence on the mass transport inside EG/GO and GO membranes. We use AFM primarily to characterize the surface roughness of EG/GO and GO membranes to shown that EG is successfully attached onto the surface of GO. As shown in the Fig. 9 in the manuscript in page 10 line 269, we can observe that the surface roughness of GO/EG membranes are significantly higher than the surface roughness of the GO membranes, which is consistent with the SEM analysis (Fig. 2).

- Table 3 is misleading and should be deleted, rejection of dyes should not be compared with rejection of other solutes, e.g., salts and proteins!

Responses: Yes, you are absolutely right, this is our mistake. Rejection of dyes should not be compared with rejection of other solutes. This is an unfair comparsion. We have deleted Table 3 in the manuscript. And we wish that all these answers will meet your requirements.

According to the comments of Reviewers, we have put some experimental results in the supporting information document in order to avoid the article being too long. At last, we express our great gratitude for reviewer’s suggestions and supporting for our work once again.

Reviewer 3 Report

The paper investigates the effect of EG on the interlayer distance of GO. This work has completed sufficient characterisation of the membrane properties and its performance. My detailed comments are below:

1) Figure 3: Please check the figure caption. Figure 3 (a).. (b).. (b)... (c)... should read (a) .. (b)... (c).. (d)..

2) Figure 5: Suggestion to Figure caption: The rejection rate of RhB of GO and GO/EG 

membranes

3) Are the results reproducible? It would be great to show error bar for Figure 4 and 5.

4) Comment on the interlayer spacing of GO/EG: Is it possible for EG to graft onto both sides of the graphene layer (form crosslinking) and therefore smaller interlayer spacing was observed?

5) Table 3: The table is not a fair comparison as the solutes in these experiments are different (in terms of size, charge and concentration).

6) Page 11, line 264: Membranes permeability and rejection test: How long was the permeability test and at what duration was the permeate sampled? We would want to avoid any possible rejection caused by absorption. Do these membranes maintain its flux and rejection after prolonged period of test? e.g. after 6 hours.

7) Page 11, line 264: Membranes permeability and rejection test: What was the operating pressure and effective area of the membrane tested?

Author Response

The paper investigates the effect of EG on the interlayer distance of GO. This work has completed sufficient characterisation of the membrane properties and its performance. My detailed comments are below:

1) Figure 3(Fig. 4 in the new version): Please check the figure caption. Figure 3 (a).. (b).. (b)... (c)... should read (a) .. (b)... (c).. (d)..

Responses: Yes, you are right, this is our mistake. And now we have revised “Figure 4 (a)... (b)... (b)... (c)...” to “Figure 2 (a)... (b)... (c)... (d)...” in the manuscript in page 7 line 199.

2) Figure 5 (Fig. 6 in the new version): Suggestion to Figure caption: The rejection rate of RhB of GO and GO/EG membranes

Responses: Thank you very much for your valuable suggestion. And now we have revised “The rejection of GO and GO/EG membranes” to “The rejection rate of RhB of GO and GO/EG membranes” in the manuscript in page 9 line 242.

3) Are the results reproducible? It would be great to show error bar for Figure 4 and 5 (Fig. 5 and Fig. 6 in the new version).

Responses: Thank you very much. Yes, the results are reproducible. We have done this experiment repeatedly before writing this article. We have also added error bars in Figure 5 and 6 in the manuscript in page 7 line 215 and in page 9 line 241.

4) Comment on the interlayer spacing of GO/EG: Is it possible for EG to graft onto both sides of the graphene layer (form crosslinking) and therefore smaller interlayer spacing was observed?

Responses: This is a very good question, the question showed that the reviewer read the manuscript in detail. First of all, we don't deny the existence of this situation you said. However, we believe that this situation can be ignored for the following reasons:

First of all, the GO/EG composites we prepared are highly hydrophilic and can be well dispersed in water. If EG to graft onto both sides of the graphene layer, this will consume a lot of oxygen-containing functional groups. Therefore, the GO/EG composite material has poor hydrophilicity and cannot be stably dispersed in water. Secondly, We can observe that the GO/EG composite material is distributed separately in the TEM image, below. If GO and EG are cross-linked, the GO/EG composite material should be seen in the TEM image as a black block. Finally, if GO and EG are cross-linked, the GO/EG can’t be filtered to form a membrane. And we wish that all these answers will meet your requirement.

Figure. R1 TEM image of the GO/EG composite material

5) Table 3: The table is not a fair comparison as the solutes in these experiments are different (in terms of size, charge and concentration).

Responses: Yes, you are absolutely right, this is our mistake. Rejection of dyes should not be compared with rejection of other solutes. This is an unfair comparsion. We have deleted Table 3 in the manuscript.

6) Page 11, line 264(Page 4, line 122 in the new version): Membranes permeability and rejection test: How long was the permeability test and at what duration was the permeate sampled? We would want to avoid any possible rejection caused by absorption. Do these membranes maintain its flux and rejection after prolonged period of test? e.g. after 6 hours.

Responses: These are very good questions. First of all, according to the equation of water permeability in the manuscript in page 4 line 127, we recorded the time of 10 ml of water passing through the membrane to calculate the water permeability. Then, different membranes have different filtration times. Therefore, the time range for the water permeability test is 4 minutes to 25 minutes. After the permeability test, we will immediately sample and do the UV test to avoid any possible rejection caused by absorption. Finally, these membranes maintain its water permeability after prolonged period of test. However, the rejection of the membranes will gradually increase after prolonged period of test due to the retentate accumulates on the surface of the membranes. And we wish that all these answers will meet your requirement.

7) Page 11, line 264 (Page 4, line 122 in the new version): Membranes permeability and rejection test: What was the operating pressure and effective area of the membrane tested?

Responses: Thank you so much for your valuable question. As shown in the manuscript in page 3 line 107, the effective diameter of each membrane was 4 cm. Therefore, we can calculate the effective area of the membrane is 1.256*10-3m2 in the manuscript in page 4 line 128. And the operating pressure of the membrane test is 1 bar (0.1 MPa) in the manuscript in page4 line 129.

Round 2

Reviewer 1 Report

The authors appear to have addressed the comments satissfactorily.

Reviewer 2 Report

The manuscript can be published in the present form.